# Astaxanthin Delivery Systems for Skin Application: A Review

**DOI:** 10.3390/md19090511

**Published:** 2021-09-09

**Authors:** Sarah Giovanna Montenegro Lima, Marjorie Caroline Liberato Cavalcanti Freire, Verônica da Silva Oliveira, Carlo Solisio, Attilio Converti, Ádley Antonini Neves de Lima

**Affiliations:** 1Department of Pharmacy, Federal University of Rio Grande do Norte, Natal 59012-570, RN, Brazil; sarahmontenegrolima@gmail.com (S.G.M.L.); veronicasoliver47@gmail.com (V.d.S.O.); 2Physics Institute of São Carlos, University of São Paulo, São Carlos 13566-590, SP, Brazil; marjorie_freire_@hotmail.com; 3Department of Civil, Chemical and Environment Engineering, Pole of Chemical Engineering, University of Genoa, I-16145 Genoa, Italy; solisio@unige.it (C.S.); converti@unige.it (A.C.)

**Keywords:** astaxanthin, delivery system, skin, cosmetics, drugs, drug release

## Abstract

Astaxanthin (AST) is a biomolecule known for its powerful antioxidant effect, which is considered of great importance in biochemical research and has great potential for application in cosmetics, as well as food products that are beneficial to human health and medicines. Unfortunately, its poor solubility in water, chemical instability, and low oral bioavailability make its applications in the cosmetic and pharmaceutical field a major challenge for the development of new products. To favor the search for alternatives to enhance and make possible the use of AST in formulations, this article aimed to review the scientific data on its application in delivery systems. The search was made in databases without time restriction, using keywords such as astaxanthin, delivery systems, skin, cosmetic, topical, and dermal. All delivery systems found, such as liposomes, particulate systems, inclusion complexes, emulsions, and films, presented peculiar advantages able to enhance AST properties, among which are stability, antioxidant potential, biological activities, and drug release. This survey showed that further studies are needed for the industrial development of new AST-containing cosmetics and topical formulations.

## 1. Introduction

Astaxanthin (AST) is a xanthophyll carotenoid that was first isolated from lobster by Kuhn and Sorensen and was commercialized as a pigmentation agent for feed in the aquatic farm industry [1,2]. The AST, or 3,3′-dihydroxy-β,β-carotene-4,4′-dione, is a tetraterpene composed of 40 carbon atoms (Figure 1A), and its molecular formula is C_40_H_52_0_4_ (molecular mass 596.85 g·mol^−1^) [3,4]. This reddish-orange pigment is solid at room temperature, is fat-soluble, and its log P (octanol/water partition) is 13.27 [4]. Furthermore, its chemical structure is composed of 13 conjugated double bonds that have the ability to neutralize free radicals, conferring the strong antioxidant activity of AST [3]. 

In recent years, AST has gained visibility and attracted attention for cosmetic and dermatological applications, thanks to its remarkable antioxidant properties, which are much stronger than those of tocopherol, and to its positive effects on skin health and protection against UV radiation, which may suggest promising applications in anti-aging products [5,6].

Regarding industrial applications, AST has been produced synthetically through cost-effective methods for large-scale production. However, the green microalgae *Haematococcus pluvialis*, due to its ability to accumulate AST at high levels, is the main source for human consumption, besides being the most promising source for its industrial biological production [1,7]. It can be found either in plants, animals, yeasts, or other algae species. Nowadays, it finds many applications in aquaculture, cosmetics, foods, nutraceuticals, and pharmaceuticals [1] (Figure 1B).

In the foodstuffs field, AST has been widely used as a supplement and healthy functional food, due to its neuroprotective properties and its ability to scavenge the free radicals produced by physical exercise, improve human immunity, resist fatigue, delay aging and prevent a number of diseases that cause organ aging. In addition, it can be used as a food colorant and antioxidant, to enhance the sensory quality and nutritional values of foods [8]. Oral consumption of AST-containing supplements seems to have positive effects on the skin, such as facial skin rejuvenation [9], an increase in flap viability [10], protection against photoaging caused by UV irradiation [11,12], and improvements in fine lines/wrinkles, elasticity, moisture, age spot size and texture [13,14].

Topical AST application has been reported to have several skin health benefits, including antioxidant and anti-aging effects [13,15,16,17,18,19], protection against UV irradiation [16,20], anti-wrinkle [14,16,21], hydration [21], wound healing [22,23], anti-cancer properties [17], and anti-eczema effects [13,24]. 

However, the low bioavailability and solubility of AST limit its use in topical formulations. The development of delivery systems to improve AST’s application for skin purposes is a promising way to develop new cosmetics and pharmaceutical products. With this goal in mind, this article aimed to review the delivery systems described in the literature for enhancing AST’s properties, discuss the main results of the developed formulations and contribute to its applications in cosmetics and topical formulations.

## 2. Astaxanthin Delivery Systems for Skin Application

### 2.1. Vesicular Systems

#### Liposomes

Liposomes are colloidal and vesicular delivery systems, composed of at least one bilayer amphiphilic lipid membrane and a hydrophilic core [25]. These systems have many advantages, including the capability of encapsulating hydrophilic, hydrophobic, or amphiphilic components, target potential, slow-release properties, and biocompatibility due to the similarity with cell membranes, as well as biodegradability, low toxicity, ease of preparation and the ability to extend product shelf-life [25,26]. The use of liposomes as a drug delivery system allows partially overcoming the problems related to the poor stability, water-solubility, and bioavailability of AST [27] (Figure 2). For skin treatment purposes, liposomes are an excellent option for drug delivery because they can be absorbed by endocytosis, particularly by cells of the reticuloendothelial system, and release entrapped drugs. They are also able to fuse with cells or exchange lipids with the cell membrane [28]. However, in aqueous solutions, liposomes have reduced vesicle flexibility, which is a problem in the preparation of efficient liposomal formulations. In addition, absorption from topical application to the skin requires that it reaches the stratum corneum, which poses a challenge for the development of new alternatives to increase efficiency. The loaded-AST liposomes studies found are summarized in Table 1**.**

Dopierała et al. [29] have recently investigated the interactions between AST and the lipid membrane, in terms of the thermodynamic, morphological, and viscoelastic behavior and surface charge density, to understand how this molecule can affect liposomal formation and characteristics. The study showed that AST can be incorporated into the lipid monolayer and form a stable film, which regulates membrane fluidity and enhances the stability of the liposomal delivery system. Through steady-state fluorescence measurements, these authors observed that the interaction between AST and the lipid membrane reduced membrane fluidity but increased its micropolarity within a certain range of AST concentration, giving more stability to the saturated lipid bilayer. This system appears to be a promising way to improve lipid-based drug formulations, designing a dosage form that may help overcome the problem of unstable bioactive compounds. In addition to AST’s antioxidant and biological activities, it is possible to enhance the stability of other formulations that are acting as adjuvants. Previously, Goto et al. [30] suggested that the two terminal rings of AST are likely to interact with the hydrophilic polar region of membrane phospholipids, and intermolecular hydrogen bonds could be formed between polar ends of the hydroxyl-ketocarotene and polar groups of phospholipids, thus regulating the fluidity of lipid membranes.

As an alternative liposomal delivery system, Lee et al. [31] developed liposomes with solid-supported lipid bilayers to overcome the flexibility issues of vesicles in an aqueous solution. Silicified liposomes were synthesized via silicification with tetraethyl orthosilicate on the hydrophilic regions of lecithin vesicles and assembled with boron nitride to enhance their stability. The resulting AST-loaded silicified-phospholipids boron nitride complex showed a release pattern and an antioxidant activity according to the 2,2-diphenyl-1-picrylhydrazyl (DPPH) free-radical scavenging method that is compatible with the literature. In addition, it presents a simple and low-cost method for AST applications in cosmetics.

However, regular liposomes have some disadvantages, due to their large particle size (around 1–100 µm). As an alternative way to enhance the use of this delivery system, Pan et al. [27] developed nanoliposomes with a lower particle size that exhibited better penetration and new targeting properties. Using a film-dispersion ultrasonic technique and soybean phosphatidylcholine as the lipid, AST-incorporating nanoliposomes (AST-LN) were prepared, characterized, and evaluated for their in vitro release, compared with the pure molecule. Nanoliposomes ensured higher encapsulation efficiency (97.5 ± 0.3%) than in previous reports, and an average particle size of 80 ± 2 nm, which was confirmed by transmission electron microscopy (TEM). AST-LN was 17 times more soluble than pure AST and had good water dispersibility, without insoluble particles or precipitation. X-ray diffraction (XRD) analyses suggested that this improvement of water solubility could be due to the change of AST’s crystalline natural form into nanoliposomes. To evaluate the release of AST-LN, an in vitro assay was carried out with pure AST solution and nanoliposomes, using a dialysis bag technique, which stopped after given intervals. The AST release from nanoliposomes after 24 h was much slower (28.74%) than from the pure solution (95.27%), which supports the potential of continuous release from nanoliposomes.

Hama et al. [18] developed liposomal topical AST formulations to assess their action against skin damage that has been induced by UV radiation. The lipid hydration method was used to prepare either neutral liposome formulations (Asx-EPC-lipo) using egg phosphatidylcholine (EPC) liposomes or 1,2-dioleoyl-3-trimethylammonium-propane-based cationic liposomal formulations (Asx-DOTAP-lipo), intended to reach the skin stratum corneum by exploiting iontophoretic delivery. The diameter of the neutral vesicles varied from 170 µm to over 300 µm, and that of the cationic ones was around 170 ± 40 µm. The in vitro test of antioxidant activity with singlet oxygen generation and chemiluminescence detection showed that liposomal formulations did not interfere with AST antioxidant activity compared with the free solution, which increased in a dose-dependent manner. The results of in vivo tests performed to protect the skin from UV radiation showed that pretreatment of the skin surface with Asx-EPC-lipo prevented morphological skin changes, which suggests that liposomal AST formulation effectively prevented stratum corneum thickening by scavenging reactive oxygen species (ROS). The effect of iontophoretic transdermal delivery with cationic Asx-DOTAP-lipo pretreatment resulted in the significant inhibition of UV-induced melanin production in the basal laminae region. This result suggests that this formulation prevented melanocyte activation by efficiently scavenging ROS in deep regions of skin exposed to UV irradiation, acting as a potential whitening agent.

In another article, Hama et al. [32] compared the hydroxyl radical scavenging activity in an aqueous solution of Asx-EPC-lipo, prepared by the method of lipid hydration using EPC as the lipid base with those of β-carotene and α-tocopherol liposomes. The average diameter of EPC-lipo increased from 150 μm to over 300 µm, alongside increasing the amount of entrapped AST. Chemiluminescence intensity, which was assessed by the Fenton reaction to check the scavenging potential of AST and its encapsulated formulations, decreased in a dose-dependent manner with respect to AST in EPC-lipo. In addition, AST-containing liposomes were more powerful than either EPC-lipo-encapsulated β-carotene or α-tocopherol. An in vitro cytotoxicity test was performed to evaluate if Asx-EPC-lipo could protect a cultured mouse skin fibroblast cell line (NIH3T3 cells) against hydroxyl radicals damage. Results showed that the formulation prevented hydroxyl radical cytotoxicity in a dose-dependent manner since liposomes could suffer from cell endocytosis, and suggested that this effect was probably due to the action on the cell surface of AST molecules distributed in the plasma membranes.

### 2.2. Emulsions

#### 2.2.1. Microemulsions

Microemulsions are thermodynamically stable transparent delivery systems formed by droplets dispersed on a liquid phase, which are capable of forming spontaneously and have low interfacial energy and size, ranging from about 10 nm to 100 nm (Figure 3A) [33,34]. Zhou et al. [35] developed AST oil/water microemulsions and checked the effect of different antioxidants either alone or in combination as additives to improve their stability. The microemulsion prepared with AST alone, with Tween 80 as an emulsifier and ethanol in buffer solution, retarded AST degradation, thereby proving to be an excellent alternative to improve emulsions for AST delivery.

#### 2.2.2. Nanoemulsions

Nanoemulsions are thermodynamically and kinetically stable systems with nanoscale droplet sizes (100 nm to 400 nm) [33], uniform size distribution, and physicochemical and biological properties different from those of other emulsions (> 500 nm) (Figure 3A) [36]. The small droplet size, the scarce probability of coalescence and flocculation, effective delivery of active ingredients, rapid penetration, long-lasting effects, and uniform deposition onto the skin make them suited for use in the personal care, cosmetic, and health science fields [37]. Articles on micro- and nanoemulsions are described in Table 2.

To obtain stabilized cosmetic products with anti-wrinkle, anti-aging, and humectant properties, Kim et al. [37] prepared oil/water AST nanoemulsions using glycerol esters such as glyceryl citrate, lactate, linoleate and oleate as alternative emulsifiers to the traditional hydrogenated lecithin. Nanoemulsions were studied in terms of their physicochemical properties, such as the emulsifier type and concentration, preparation conditions, and AST concentration, and were characterized by a freeze-fracture scanning electron microscope (FF-SEM), TEM, and high-performance liquid chromatography (HPLC). Nanoemulsions, with zeta potential in the range from −10 to −57 mV and an average particle size of 170 nm, were progressively more unstable with the increase in particle size when hydrogenated lecithin was used.

Hong et al. [38] prepared AST nanoemulsions that were functionalized with carboxymethyl chitosan, to investigate its effects on the droplet size, stability, skin permeability and cytotoxicity of the formulation. For this purpose, the low-energy emulsion phase inversion method was used, which, in addition to preventing AST degradation during nanoemulsion preparation, has the advantages of a low cost, high energy efficiency, simplicity of production and easy scale-up. Results showed improvement in chemical stability and skin permeability, while the small droplet size, satisfactory physical stability, and low cytotoxicity were not affected by the functionalized nanoemulsion.

In a comparative study, Shanmugapriya et al. [39] prepared uniform and stable oil/water nanoemulsions containing AST or α-tocopherol by the spontaneous and ultrasonication emulsification methods, using an experimental design for optimization. The AST-containing nanoemulsions displayed anticancer, wound healing and antimicrobial effects, which suggests their use in formulations for the treatment of skin cancer and wound healing, for example via incorporation into films. The same research group [39] used AST and α-tocopherol nanoemulsions with κ-carrageenan to verify topical wound healing effects in vitro and in vivo. Results suggested the treatment as a good alternative for wounds in diabetic cases with higher activity in a shorter time.

### 2.3. Particulate Systems 

Particulate systems are delivery systems that include nano/microspheres and nano/microcapsules (Figure 3B) [25]. Whereas nano/microspheres are dispersions of active ingredients in polymeric matrices, nano/microcapsules are reservoirs where distinct domains of core and wall material are present. These formulations create a compatible environment for susceptible molecules and protect them from light, oxygen, pH, heat, enzymatic degradation and other external factors that can affect their stability [41]. Microencapsulation technologies are widely used in cosmetic products to increase stability, protect against degradation, promote safe administration and allow controlled and targeted release [42]. Particle size varies from around one micron to a few millimeters, providing a large surface area available to develop active ingredients. Nanoparticles involve particulate systems in the nano range, providing a larger surface area that is available for adsorption and desorption sites, chemical reactions, light scattering, etc. [42]. Studies on which particulate systems have been used are summarized in Table 3.

#### 2.3.1. Microparticles

Higuera-Ciapara et al. [44] developed non-homogeneous-sized AST microcapsules with a diameter of 5–50 µm, using a chitosan matrix cross-linked with glutaraldehyde by the multiple emulsion/solvent evaporation method. The system stability was evaluated based on retained pigment quantity during microcapsule storage at 25, 35 and 45 °C, which was quantified weekly by HPLC. For any of the treatments, the results did not show a marked decrease in AST concentration in the microcapsules, and AST was maintained in stable conditions, as evidenced by poor isomerization and pigment degradation.

Oil bodies (OB) are minute plant organelles similar to liposomes, with 0.5–2.0 µm diameter, consisting of an oil core surrounded by a phospholipid monolayer with a proteinaceous membrane, and these are used to deliver phytohormones and other hydrophobic compounds in plants. These structures, which have been isolated from rapeseeds, have been shown to constitute a novel type of microcapsule that is suitable for the extraction of hydrophobic organic compounds from aqueous environments [52]. Hydrophobic compounds like AST that are used in cosmetics have been loaded into these systems, to protect them from oxidation. Acevedo et al. [43] developed AST microcapsules (AST-M) with OB extracted from *Brassica napus* seeds with high microencapsulation efficiency (>99%) and used a response surface methodology to optimize the microencapsulation conditions. The AST-M were examined by optical microscopy, which evidenced morphological stability, autofluorescence, spherical structures and an AST presence in the core. Their larger mean diameter (3.4 ± 0.5 mm) compared to those containing OB alone (1.56 ± 0.06 mm) was likely due to the AST intercalation into microcapsule monolayer, as previously described by other authors. Stability studies showed high stability of microcapsules to aggregation and coalescence, as well as a double half-life in the presence of air and light exposure compared with free AST, thereby highlighting the protective role of OB against AST degradation. After 2 h of cell incubation (CRL1730 endothelial cell line), antioxidant assays demonstrated the higher antioxidant power of AST-M in comparison with free AST, which was dose- and time-dependent. The cell viability assay did not show any cell toxicity, and microencapsulated AST displayed higher oxidative stability than its free counterpart. The authors suggested that the use of OB as a new delivery system is promising for the cosmetic field because it joins once in contact with the skin, releasing the antioxidant safely, and offers a new and natural carrier to deliver stable AST.

Lin et al. [45] searched for the best conditions to prepare AST that is encapsulated in sodium alginate beads, varying the concentrations of calcium chloride solution, medium, sodium alginate solution as the encapsulation agent, and Tween 20 as a surfactant, and selecting the average yield weight, microencapsulation yield, the average size of beads and loading efficiency as the responses. It seemed that the higher the alginate concentration, the higher the average weight and size of the beads. Different conditions of three-week storage were tested for free AST, AST encapsulated in various formulations, and a commercial product with 10% AST, in terms of bioactive content. The percentage of AST amount retained after storage was higher (90%) for beads compared to the controls without encapsulation, which suggests that the matrix covering the molecule protected it from thermal degradation and oxygen attack.

AST microspheres were developed by Liu et al. [46] using the supercritical anti-solvent (SAS) process and poly (L-lactic acid) as the polymeric carrier. The authors believe that this alternative method has great potential for preparing encapsulation systems, thanks to its single-pass process and mild operating conditions, to prevent the degradation of AST and other sensitive molecules. SAS operating conditions were varied according to an orthogonal experimental design to elucidate encapsulation circumstances. Optimal conditions ensured an encapsulation efficiency of 91.5% and a mean particle size of 954.6 nm. Characterization assays proved the formation of uniform particles and the amorphous state of AST encapsulated in the matrix, while 6-month storage tests at 40 °C showed the enhancement of its stability.

#### 2.3.2. Nanoparticles 

Poly(lactic-co-glycolic acid) (PLGA) is a copolymer recently proposed for use in topical formulations to prevent and treat photodamage, one that is non-toxic and can be hydrolyzed in vivo into its biodegradable monomers, i.e., lactic and glycolic acids [53]. PLGA-AST nanoparticles (AST-PLGA NP) were developed by Hu et al. [47] in an optimization study, based on an experimental design, aiming to maximize encapsulation efficiency (96.42 ± 0.73%) and drug-loading capacity (7.19 ± 0.12%), and to simultaneously minimize particle size (154.4 ± 0.35 nm). Fourier transform infrared (FT-IR) spectroscopy, differential scanning calorimetry and thermogravimetric analyses confirmed AST encapsulation within PLGA nanoparticles, while XRD analysis suggested its amorphous state, and scanning electron microscopy (SEM) and TEM evidenced globular nanoparticles without drug crystals. This would contribute to easier penetration compared to pure AST, and would afford good size distribution. To evaluate the capacity of cellular uptake, an in vitro model with a fluorescent probe was performed with HaCaT cells, an aneuploid immortal keratinocyte cell line. The results showed cellular uptake of the nanoparticles that increased in a time-dependent way. In vitro cytotoxicity, assessed by the MTT assay, showed no significant decrease in cell viability, while AST-PLGA NP increased cell viability and had stronger antioxidant activity, compared to pure AST. Protection by scavenging free radicals was performed on the same cell line, and pure AST and AST-PLGA NP displayed similar antioxidant properties.

Chitosan nanoparticles are promising as drug delivery systems, due to their biocompatibility, biodegradability, atoxicity, bioactivity, and large target options triggered by their cationic character [54]. Chitosan and natural DNA were used by Wang et al. [49] as oppositely charged biomaterials, to develop a nanoparticle system to deliver and release AST. Thus, AST-loaded DNA/chitosan (ADC) and empty nanocarrier (DNA/CS) were prepared by macromolecular co-assembly combined with the solvent evaporation method, and their antioxidant activities and cellular uptake were assessed. Both showed a positive surface charge, due to the predominant cationic characteristic of chitosan, and very close zeta potential (35.3 and 32.2 mV, respectively), which suggests that AST incorporation did not influence this parameter. The larger average particle size of ADC (211 ± 5 nm) compared to DNA/CS, and its PDI of 0.29 ± 0.01, confirmed its nano scale, good dispersity, and AST entrapment. The AST content in ADC was higher than in ethanol solution, confirming enhancement of its solubility. The MTT assay, performed to assess the cytoprotective effect, showed that ADC prevented the deleterious effect of H_2_O_2_ on cell viability more than vitamin C (positive control) and free AST, in spite of a vitamin C concentration that was three times higher than that of AST, likely due to the facilitated endocytosis of ADC nanoparticles. The ROS scavenging efficiency of ADC nanoparticles was twice that of free AST at the same bioactive concentration.

Liu et al. [50] prepared an innovative type of nanoparticle to improve solubility and stability, using chitosan oligosaccharides (COS) to coat PLGA AST nanoparticles (Ax-PLGA@COS NP). To prepare AST-loaded PLGA nanoparticles, a two-step process was used, which consisted of antisolvent precipitation, followed by coating nanoparticles with COS by electrostatic deposition. The formulation was characterized in terms of encapsulation efficiency, morphology, PDI, zeta potential and particle size, while the stability of nanoparticles was evaluated in terms of changes in their color and particle size during 72 h of storage at room temperature. Ax-PLGA@COS NP suspensions had a uniform orange color that did not change appreciably during storage. Furthermore, no particle aggregation or sedimentation was observed in these systems, and the formulation showed no cytotoxicity to Caco-2 cells.

Due to AST’s susceptibility to heat, Tachaprutinun et al. [51] aimed to assess the resistance to thermal degradation of AST nanoparticles prepared by the solvent displacement process, using three different polymers, namely poly(ethylene oxide)-4-methoxy cinnamoyl phthaloyl-chitosan (PCPLC), poly(vinylalcohol-co-vinyl-4-methoxycinnamate) (PB4) and ethylcellulose (EC). While EC was inefficient and PB4 poorly efficient for encapsulating the bioactive compound, PCPLC allowed for an encapsulation efficiency of 98%, a loading of ~40%, and a 312 ± 5.83 nm particle size. In thermal degradation tests performed at 70 °C for two hours, most of the free AST was degraded, while the nanoparticle formulation was able to protect the molecule from degradation.

Solid lipid nanoparticles (SLN) are colloidal particles prepared from solid lipids, surfactants, the active ingredient and water. This method has shown several advantages, including the use of biocompatible lipids, high in vivo stability and a wide application spectrum. However, they have limitations, such as a low drug-carrying capacity and drug leakage during storage. Thus, nanostructured lipid carriers (NLC) emerge as second-generation lipid nanoparticles to overcome these deficiencies [55].

NLC are obtained using lipids (solids and liquids) and emulsifiers, forming a solid lipid matrix at body temperature that is capable of incorporating hydrophobic and hydrophilic molecules [25]. This matrix has promising potential for the pharmaceutical and cosmetics industry due to its beneficial effects, such as skin hydration, occlusion, greater bioavailability, and targeting application to the skin [55].

In this sense, Rodriguez-Ruiz et al. [48] developed an innovative possible solution with a green chemistry process formulation of AST-loaded nanostructured lipid carriers (NLC). The compounds used to synthesize NLC by the hot homogenization method were sunflower oil as the liquid lipid, glyceryl palmitostearate as the solid lipid, and Poloxamer 407 and Tween 80 as the surfactants. AST-loaded NLC and AST-free NLC were characterized by dynamic light scattering (DLS), atomic force microscopy (AFM) and SEM. DLS analysis showed that for the former particle size of 60 ± 7 nm, a polydispersity index (PDI) of around 0.33 ± 0.09 and zeta potential of −25.5 ± 0.7 mV were achieved, which is indicative of stability. The AFM and SEM analyses demonstrated nanoparticle spherical shape and nanoscale. To evaluate the stability, samples were stored for one month, protected from light, at low temperatures. Results showed no significant change in the size, PDI and zeta potential of both preparations. No less than 90 ± 5% of the starting AST content in NLC was observed after this time, and the additional effect of antioxidants present in sunflower oil seemed to protect the bioactive compound from oxidation. The antioxidant potential determined by the α-tocopherol equivalent antioxidant capacity assay demonstrated an inhibition curve slope that was almost double for AST-loaded NLC compared to the control, confirming the high antioxidant activity of AST [48].

### 2.4. Inclusion Complexes

#### Cyclodextrin

Cyclodextrins (CDs) are a family of cyclic polysaccharides used to form inclusion complexes with a wide variety of substances used in pharmaceuticals, drug delivery systems, cosmetics, and in the food and chemical industries [56]. Their molecular structure is composed of a cavity size, which is determined by the number of glucose units, where the space inside the cyclodextrin molecules allows the formation of inclusion complexes with poorly soluble compounds (Figure 3C) [56]. The inclusion of guest molecules into CDs can change their physical and chemical properties, as well as increasing their water solubility and stability [57]. The CDs are an excellent alternative for the inclusion of a variety of natural compounds, such as oils [58,59] and other compounds [60,61]. The AST inclusion complexes with cyclodextrin are summarized in Table 4.

One of the first studies on the inclusion of AST in CDs to enhance its solubility for topical applications was developed by Lockwood et al. [62]. When used in proportions from 0 to 60% (*w/v*), a sulfobutyl ether β-cyclodextrin was shown to complex with crystalline AST. At 60%, AST water solubility has increased by more than 50 times, and the implementation of a pre-solubilization process could increase it by 71 times over the parent compound in water.

Kim et al. [63] prepared AST inclusion complexes with various types of CDs in different ratios, characterized each formulation by HPLC, SEM and FT-IR, and evaluated their 28-day stability and water solubility under different conditions of pH, light, temperature and oxidation. To minimize the costs, a β-cyclodextrin (β-CD) that is widely used in food and cosmetics applications was used for comparison. Inclusion complexes formed at the AST ratio of 1:200, and the host molecule showed a uniform shape and particle size. β-CD was proved to incorporate AST, with an inclusion yield higher than 90%, and the solubility of the resulting AST-loaded inclusion complex was 13-fold at 25 °C and about 100-fold that of free AST at a pH of 6.5. In the stability study, the yield of the inclusion complex remained above 80% after 21 days of UV irradiation, while the free AST was completely degraded. In addition, it proved stable against oxidation, was favored by acidic conditions and exhibited greater temperature resistance for industrial processing.

Chen et al. [57] prepared an AST β-cyclodextrin complex and measured its water solubility and stability to heat and light. Complex formation was checked by infrared spectroscopy and HPLC, showing an inclusion yield of 48.96%. The water solubility of AST was slightly increased, while its heat stability was greatly enhanced compared to the free bioactive compound.

Hydroxypropyl-β-cyclodextrin (HP-β-CD) is a hydroxyalkyl derivative alternative to parent CDs, which offers improved water solubility and is slightly more friendly from a toxicological standpoint [68]. Yuan et al. [64] prepared a new water-soluble formulation of AST with HP-β-CD, analyzed its thermal behavior, and investigated its stability in heat and light. The inclusion yield of the formulation was 46.5%, and no less than 200 mg/mL of it was dispersed in water. Water solubility was greatly enhanced (>1.0 mg/mL) in comparison with the previous study, due to the higher solubility of HP-β-CD compared with β-CD. On the other hand, the overall amount of AST entrapped in the inclusion complex was lower, likely because the hydroxypropyl substituent concentrated at the edge of the CD cavity made the entry of AST molecules more difficult. In 2012, the same research group studied this complex via UV-Vis, FT-IR, 1H nuclear magnetic resonance (NMR) spectroscopies and molecular modeling, to enhance knowledge about the molecule structure [65]. Storage stability at 4 and 25 °C was higher than that of free AST, while in vitro antioxidant tests showed greater antioxidant activity than ascorbic acid [66].

To prevent antioxidative stress on endothelial cells, Zuluaga et al. [67] developed a similar complex with AST and HP-β-CD. The differential of this study was the direct and indirect measurement of its antioxidant capacity by understanding the cells’ molecular mechanisms involved in gene expression. Results showed that the inclusion complex formulation could protect cells by activating endogenous AST systems through the Nrf2/HO-1/NQO1 pathway, in addition to the enhancement of solubility due to its incorporation into the cyclodextrin core.

### 2.5. Films

Topical film-forming systems are drug delivery systems for topical applications that are capable of adhering to the skin, forming a thin transparent film that provides delivery of the active ingredients to the body tissue (Figure 3D) [69]. These systems are composed of the drug- and film-forming agents in a volatile vehicle, which evaporates in contact with the skin [69]. Even though it is a promising option for topical drug delivery, the literature on the incorporation of AST as a way to exert antioxidant effects on the skin is scarce.

Veeruraj et al. [23] prepared films to demonstrate the wound healing properties of AST when incorporated in collagen films. In vivo assays were performed to assess tissue regeneration and drug delivery from the formulation, and in vitro assays to check the antioxidant activity. In addition to the AST collagen film, a gentamicin-incorporating collagen film was developed to assess its antibiotic effects. The filming agent and AST were extracted from the waste material of the outer skin of the squid *Doryteuthis singhalensis*, which is an innovative and sustainable alternative for the development of this delivery system. Biodegradation tests showed that film materials degraded more rapidly than the collagen matrix, suggesting the controlled degradation of collagen materials. Wound healing activity was measured by the reduction of the non-healing area in the healing process, occurring over 21 days. The untreated control exhibited the lowest wound contraction, whereas the AST collagen film showed the highest one among the experimental groups, as well as the fastest wound-healing progress, with complete healing in 15 days. Antioxidant assay by the DPPH free radical scavenging method showed the higher activity of AST collagen film compared to ascorbic acid. The main information on this article is presented in Table 5.

## 3. Discussion

Liposomes in themselves are relatively unstable delivery systems because of their membrane instability in aqueous solution, which can affect their bioavailability and pharmacological effect, the addition of adjuvants being necessary to bring more rigidity and stability. AST seems to enhance the liposomes’ stability when on the membrane, which helps its delivery. The strategy of adding other adjuvants to improve membrane rigidity seems to be necessary to ensure stability, which should be taken into account when assessing production costs. Reducing the particle size with the development of nanoliposomes seems to be an excellent alternative method to improve their stability and consequently their penetration, solubility, and continuous release; however, it is still necessary to consider the machinery and costs involved. Another innovative alternative method to deliver AST on deep skin layers is the use of the iontophoretic technique with charged delivery systems, developed with liposomes, which makes it possible to reach the stratum corneum and act as a whitening agent.

Emulsion delivery systems, especially nanoemulsions, are an effective and widely applied form to deliver bioactive compounds in medicines and cosmetics. Due to the hydrophobic character of AST, most of the nanoemulsions prepared in the literature are oil/water systems. Studies based on experimental design allow researchers to identify the best conditions for nanoemulsion preparation, with reduced time and costs of research. In addition, nanoemulsions can be incorporated in other delivery systems, such as films, which can significantly improve their biological effects and their acceptance by the patient or customer. Unfortunately, only one article was found concerning the use of microemulsions in dermatological and cosmetics applications. More investment in research on this new product delivery system is desirable.

Particulate systems are a promising alternative way to develop innovative formulations for AST delivery, thanks to the possibility of using different matrix components and methods to form spheres and capsules at micro- or nanoscale. Similar to nanoemulsions, studies based on experimental design can be performed to optimize the conditions needed to prepare particulate systems. It is important to consider alternative methods, to prevent AST degradation. Some articles on particulate systems considered in this review presented either ecological or natural alternatives to develop delivery systems, which can aggregate to more sustainable products. 

The use of CDs as systems to guest AST in inclusion complexes is able to significantly improve the solubility of hydrophobic compounds like AST. However, it would be necessary to have a greater number of inclusion complexes, in order to carry out a comparison by which to identify the most advantageous system. From the results examined in this review, it is possible to state that AST can form inclusion complexes either with natural CDs, such as β-CDs, or with modified CDs, such as hydroxypropyl β-CD. Films for the delivery of bioactive compounds on the skin are an alternative, especially for wound healing and scald treatments. The antioxidant effect accelerates the healing process, due to its anti-inflammatory activity. The AST-incorporating films may also be applied as an easy adjunct treatment, to treat skin cancer, which should be considered in the research. In addition, AST seems to act merely as an active principle, which requires a film-forming agent such as collagen, chitosan, cellulose derivatives, and others to create the film structure.

The systems that seem to be more developed and robust are the emulsions and the particulate systems, given the large number of articles and assays performed on them. However, they are not the only systems to consider. On the other hand, the limited studies carried out on other systems, many of which are not mentioned in this review article, show opportunities for the development of new and innovative formulations. Examples are solid dispersions, transdermal systems, and others. 

## 4. Conclusions

The aim of this article was to review the literature about the development of new systems for loading AST for cosmetics and topical usage. Delivery systems are useful for the improvement of the physicochemical profile of this compound, such as stability, water-solubility, antioxidant properties, drug release, and in vivo and in vitro biological activities. Therefore, the delivery systems for loading AST described in this article create opportunities for industrial applications; however, other industrial issues for the development of new products must be evaluated. Moreover, AST-optimized products become potentially attractive for the development of further studies, due to the antioxidant properties and benefits of AST.

## Figures and Tables

**Figure 1 marinedrugs-19-00511-f001:**
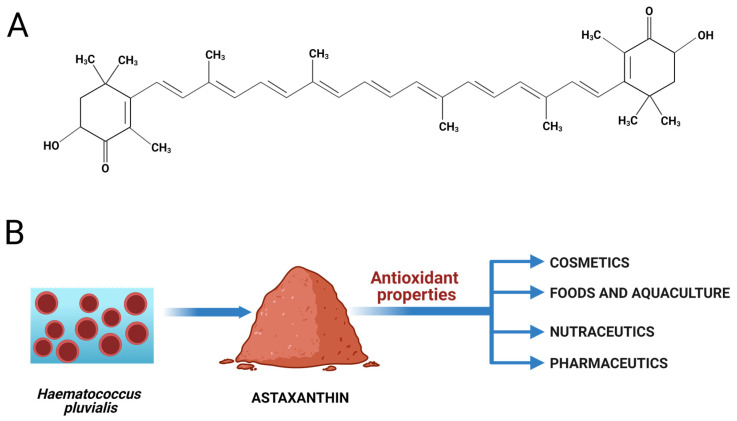
(**A**) Astaxanthin (AST) chemical structure. (**B**) The microalgae *Haematococcus pluvialis* is a promising source for AST industrial biological production and AST applications. Diagram created using BioRender.com (accessed on 17 August 2021).

**Figure 2 marinedrugs-19-00511-f002:**
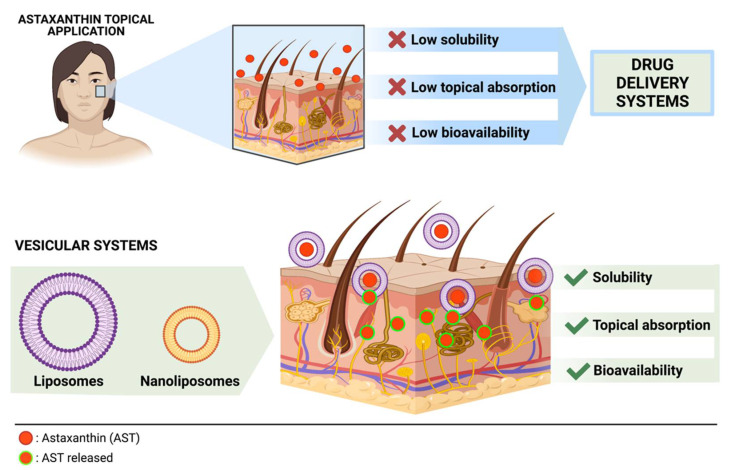
Disadvantages of the topical administration of AST and the use of delivery systems to improve the properties of the molecule, favoring skin absorption. Below: a schematic representation of vesicular systems (liposomes and nanoliposomes) acting as delivery systems for AST. Created with BioRender.com (accessed on 17 August 2021).

**Figure 3 marinedrugs-19-00511-f003:**
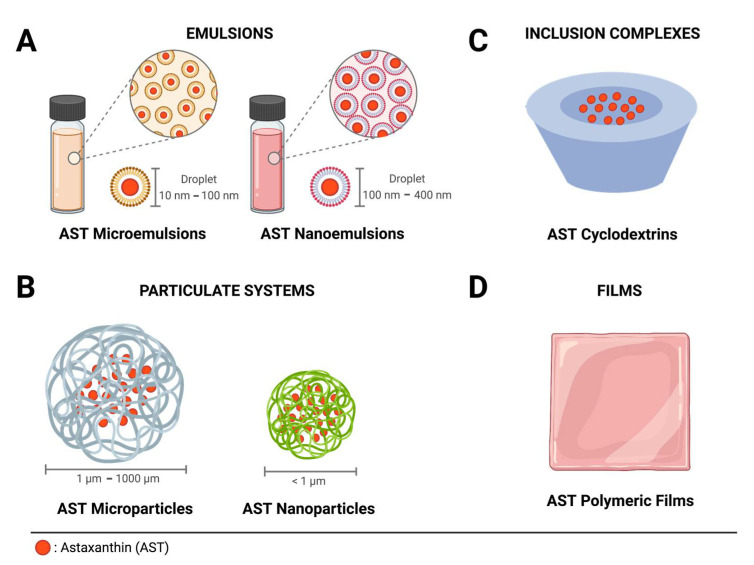
Schematic representation of AST delivery systems. (**A**) Emulsions: AST microemulsions and nanoemulsions; (**B**) particulate systems: AST microparticles and nanoparticles; (**C**) inclusion complexes: AST cyclodextrins (CDs); (**D**) films: AST polymeric films. Created with BioRender.com (accessed on 17 August 2021).

**Table 1 marinedrugs-19-00511-t001:** Summary of AST vesicular delivery systems, describing the preparation technique, liposome type, characterization and stability data, and assays (in vitro/in vivo) that were performed for each system.

Preparation Technique	Liposome Type	Characterization	Storage and Stability Data	Assays	References
Dissolution of hydrogenated lecithin and treatment by a high-pressure homogenizer to form nanoemulsions and tetraethyl orthosilicate addition to promote silification	Lecithin silicified liposomes	Brauner–Emmett–Teller isotherm, field emission scanning electron microscopy, Fourier transform infrared spectroscopy, UV–visible spectrophotometry	-	In vitro: DPPH free radical scavenging activity and drug release profile	[31]
Film dispersion-ultrasonic technique	Soybean phosphatidylcholine nanoliposomes	Dynamic light scattering, transmission electron microscopy, X-ray diffraction, differential scanning calorimetry, thermal gravimetric analysis, and dissolution study.	Thermal stability enhanced after encapsulation	In vitro: drug release profile	[27]
Lipid hydration method	Egg phosphatidylcholine liposomes	Dynamic light scattering	-	In vitro: antioxidant activity In vivo: UV treatment of mouse dorsal skin and effect of iontophoretic transdermal delivery	[18]
Lipid hydration method	Egg phosphatidylcholine liposomes	-	-	In vitro antioxidant activity by scavenging hydroxyl radical, and protective effect against cytotoxicity induced by hydroxyl radical	[32]

**Table 2 marinedrugs-19-00511-t002:** Summary of AST emulsions delivery systems, describing the preparation technique, emulsion type, characterization and stability data, and assays (in vitro/in vivo) that were performed for each system.

Preparation Technique	Emulsion Type	Characterization	Storage and Stability Data	Assays (In Vitro, In Vivo)	References
High-pressure homogenization	Oil/water nanoemulsion, glyceryl ester and hydrogenated lecithin as emulsifiers	Dynamic light scattering and transmission electron microscopy	Stability maintained for one month of storage	-	[37]
Low-energy emulsion phase inversion method	Oil/water nanoemulsion functionalized carboxymethyl chitosan	Droplet size, zeta potential and transmission electron microscopy	Stability without alteration for three months	In vitro: skin permeation studies, Cell viability assays on L929 cells, Cell culture and cytotoxicity assays	[38]
Spontaneous and ultrasonication emulsification methods	Oil/water nanoemulsion	Dynamic light scattering and transmission electron microscopy	Interference of storage conditions	In vitro: cytotoxicity (MTT assay), antimicrobial activity and scratch wound healing assay	[39]
Spontaneous and ultrasonication emulsification methods	Oil/water nanoemulsion	Dynamic light scattering and transmission electron microscopy, Fourier transform infrared spectroscopy, differential scanning calorimetry, X-ray diffraction, thermal gravimetric analysis, and scanning electron microscopy	-	In vitro: cytotoxicity (MTT assay), scratch wound-healing assay. In vivo: wound healing in nondiabetic and diabetic mice	[40]
Oil phase dispersed with AST in ethyl butyrate and homogenizing with aqueous phase in a high-speed blender and high-pressure microfluidizer	Oil/water microemulsions	Dynamic light scattering and UV-visible spectrophotometry	-	-	[35]

**Table 3 marinedrugs-19-00511-t003:** Summary of AST particulate delivery systems, describing the preparation technique, system type, characterization and stability data, and assays (in vitro/in vivo) that were performed for each system.

Preparation Technique	System Type	Characterization	Storage and Stability Data	Assays	References
AST microencapsulation by response surface methodology	Oil bodies (isolated from mature seeds) microcapsules	Fourier transform infrared spectroscopy (FT-IR), flow cytometry and microscopy	Oxidative stability, double half-life compared to free AST	In vitro: absorption assay	[43]
Multiple emulsion/ solvent evaporation	Chitosan matrix cross-linked with glutaraldehyde microparticles	AST extract analysis by high-performance liquid chromatography (HPLC)	Pigment quantity during microcapsules storage at 25, 35 and 45 °C	In vitro: storage stability evaluation	[44]
Extrusion	Calcium alginate microparticles	Analysis of AST content by HPLC	Various environmental conditions: light, temperature and nitrogen gas	In vitro: assay of AST content	[45]
Supercritical anti-solvent	Poly(L-lactic acid) microspheres	Scanning electron microscopy (SEM), transmission electron microscopy (TEM), FT-IR, X-ray diffraction (XRD), thermal gravimetric analysis (TGA), differential scanning calorimetry (DSC), UV-visible spectrophotometry	6-Month measurements by UV–vis spectrophotometry	In vitro: assay of AST content and AST release profile	[46]
Emulsion solvent evaporation	Poly(lactic-co-glycolic acid) (PLGA) copolymer nanoparticles	Dynamic light scattering (DLS), SEM, TEM, FT-IR, XRD, TGA, DSC	-	In vitro: anti-photodamage effect in HaCaT cells	[47]
Hot homogenization	Nanostructured lipid carriers	DLS, atomic force microscopy, SEM	Samples stored at 4 °C, protected from light for 1 month	In vitro: antioxidant activity by the α-tocopherol equivalent antioxidant capacity assay	[48]
Macromolecular co-assembly combined with solvent evaporation	Natural DNA and chitosan nanocarriers	DLS, TEM, field emission SEM, HPLC (AST content)	-	In vitro: oxidative stress, cytotoxicity (MTT assay) and cell uptake assay	[49]
Antisolvent precipitation method combined with electrostatic deposition method	PLGA and chitosan oligosaccharides nanoparticles	DLS, SEM, TEM, FT-IR, XRD, DSC	72 h of storage at room temperature	In vitro: cytotoxicity and AST release profile	[50]
Solvent displacement process	Ethylcellulose, Poly(ethylene oxide) 4-methoycinnamoyl-phthaloylchitosan and poly(vinylalcohol-covinyl-4- methoxycinnamate nanospheres	SEM, TEM	Thermal stability	In vitro: AST release profile	[51]

**Table 4 marinedrugs-19-00511-t004:** Summary of AST cyclodextrins (CDs) delivery systems, describing the preparation characterization, storage and stability data, and assays (in vitro/in vivo) that were performed for each system.

CDs	Characterization	Storage and Stability Data	Assays	References
β-cyclodextrin (β-CD)	High-performance liquid chromatography (HPLC), scanning electron microscopy and Fourier transform infrared spectroscopy (FT-IR)	Stability enhanced by over 7–9 folds under various storage conditions such as pH, temperature, ultraviolet irradiation, and presence of oxygen	In vitro: water solubility	[63]
Sulfobutyl ether β-CD	UV-visible spectrophotometry	-	In vitro: water solubility	[62]
β-CD	HPLC	Storage at 4, 30, 57 °C and under light (light intensity of 1500 lux)	In vitro water solubility	[57]
Hydroxypropyl- β-cyclodextrin (HP-β-CD)	Thermogravimetry, UV-visible spectrophotometry, FT-IR, molecular modeling, nucleic magnetic resonance	Stability under oxygen and light at 4, 25 and 50 °C, storage at 4 and 25 °C in dark incubators	In vitro: water solubility, antioxidant capacity by reducing power, DPPH free radical scavenging activity and hydroxyl radical scavenging activity	[64,65,66]
HP-β-CD	FT-IR, UV-visible spectrophotometry	Storage at 6 °C under light protection for 6 months	In vitro cytoprotective activity of HP-β-CD complex. Direct biological evaluation of HP-β-CD antioxidant capacity Indirect HP-β-CD antioxidant protection against reactive oxygen species	[67]

**Table 5 marinedrugs-19-00511-t005:** Summary of AST film delivery systems, describing the preparation technique, filming agent, characterization and stability data, and assays (in vitro/in vivo) that were performed for the system.

Preparation Technique	Filming Agent	Characterization	Storage and Stability Data	Assays
Collagen solution incorporating AST and gentamicin	Biomaterials extracted from the waste material of the outer skin of the squid *Doryteuthis singhalensis*	Scanning electron microscopy, energy dispersive X-ray spectroscopy, X-ray diffraction	-	In vitro: biodegradation study and DPPH free radical scavenging activity In vivo: wound-healing activity

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
