# Peer review of "Astaxanthin Delivery Systems for Skin Application: A Review"

_marinedrugs, 2021, doi:10.3390/md19090511_

Round 1

Reviewer 1 Report

It's a very interesting work related to skin research. I think it deserved to be published in marine drugs. 

Author Response

Answer: We would like to thank you for reviewing the manuscript and for your acceptance.

Reviewer 2 Report

It is a well organized review with many significant  informations regarding the delivery systems of astaxanthin.

I think that some details regarding the physicochemical behavior of astaxanthin i.e. Log P calculated and experimental (octanol/water) in order to understand better the selected encapsulation systems could add to the scientific interest of the manuscript

Author Response

Answer: We would like to thank you for reviewing the manuscript and for your suggestion. The details regarding physicochemical aspects were included in the second paragraph of introduction section (lines 30 to 35, highlighted). In addition to the log P, we inserted additional information about chemical name, molecular formula, molecular weight and some structural features that are related to the antioxidant function of astaxanthin.

Reviewer 3 Report

In the present review article, the authors present an extensive overview of the systems used to formulate astaxanthin for delivery.

It is a nice overview of the systems used in astaxanthin delivery.

I have only minor comments

  1. I suggest that the authors in the introduction section line 33 or 40 should cite the following paper on repositioning compounds as drugs which is new idea that incorporates astaxanthin (initially used as food or antioxidant and now it repositioned for cosmetics)

Sotiropoulou et al. (2021) Redirecting drug repositioning to discover innovative cosmeceuticals. Exp Dermatol 30: 628-644.

  1. I think that the authors need slightly to expand the paragraph on NLC. It appears that NLC are type of emulsions. The differences should be highlighted.

Author Response

  1. I suggest that the authors in the introduction section line 33 or 40 should cite the following paper on repositioning compounds as drugs which is new idea that incorporates astaxanthin (initially used as food or antioxidant and now it repositioned for cosmetics)

Sotiropoulou et al. (2021) Redirecting drug repositioning to discover innovative cosmeceuticals. Exp Dermatol 30: 628-644.

Answer: We would like to thank you for reviewing the manuscript and for your suggestion. The suggested reference was cited in the introductory section at the end of second paragraph (line 40). Certainly, this reference enhanced this section. This insertion is highlighted in the text (reference number 6).

  1. I think that the authors need slightly to expand the paragraph on NLC. It appears that NLC are type of emulsions. The differences should be highlighted.

Answer: Again, we would like to thank you for suggestions. The paragraph that mentions about NLC was moved to the end of the nanoparticles section. We chose to remain with the paragraph in the same section, considering NLC as lipid nanoparticles, as referenced in the article: “Chauhan, I.; Yasir, M.; Verma, M.; Singh, A.P. Nanostructured lipid carriers: A groundbreaking approach for transdermal drug delivery. Adv. Pharm. Bull. 2020, 10, 150”. Additionally, a brief introduction to NLC was inserted, reporting them as alternatives to meet the needs of first-generation nanoparticles (Solid Lipid Nanoparticles), as described in lines 364-375 (highlighted).
